# Design of Mutual-Information-Maximizing Quantized Shuffled Min-Sum Decoder for Rate-Compatible Quasi-Cyclic LDPC Codes

Peng Kang [1] , Kui Cai [1,*] and Xuan He [2]

1   Science, Mathematics and Technology (SMT) Cluster, Singapore University of Technology and Design (SUTD), Singapore 487372, Singapore
2   School of Information Science and Technology, Southwest Jiaotong University, Chengdu 611756, China
*   Correspondence: cai_kui@sutd.edu.sg

**Abstract:** In this paper, we propose a finite alphabet iterative decoder (FAID) named rate-compatible mutual-information-maximizing quantized shuffled min-sum (RC-MIM-QSMS) decoder, for decoding quasi-cyclic low-density parity-check (QC-LDPC) codes with various code rates. Our proposed decoder exchanges the coarsely quantized messages represented by symbols from finite alphabets and adopts single-input lookup tables (LUTs) to implement the node updates. To construct the LUTs used for decoding, we first propose a modified density evolution by considering the shuffled schedule to generate the LUTs which vary with different layers and iterations. Furthermore, to reduce the memory requirement for storing the LUTs, we optimize the constructed LUTs into a unique set of LUTs that only change with different decoding iterations. To the best of our knowledge, the RC-MIM-QSMS decoder is the first one to integrate the rate compatibility of LDPC codes with the shuffled decoding schedule. Simulation results show that the proposed RC-MIM-QSMS decoder outperforms the floating-point shuffled belief propagation decoder in the high signal-to-noise region and achieves comparable convergence speed to other state-of-the-art FAIDs. Moreover, the RC-MIM-QSMS decoder is able to save up to 93.22% memory requirement compared to the benchmark MIM-FAIDs.

**Keywords:** low-density parity-check codes; finite alphabet iterative decoder; mutual-information-maximizing; quasi-cyclic; rate-compatible

## 1. Introduction

Low-density parity-check (LDPC) codes [1] have been widely applied to diverse applications, such as wireless communication and data storage systems [2,3], due to their capability of approaching the capacity under iterative message passing decoding [4]. Many researchers have devoted to developing efficient LDPC decoders [5–7] to achieve a trade-off between the error rate performance and decoding complexity.

Recently, a class of finite alphabet iterative decoders (FAIDs) [8–16] have drawn much attention due to their excellent performance by using coarsely quantized messages. Due to the use of messages quantized by a low bit width, these FAIDs also achieve a low decoding complexity and are in favor of services and applications such as the Internet of things [17] and wireless sensor networks [18–20], which require strict power constraint for the devices. Different from the conventional LDPC decoders such as [5], these FAIDs exchange the messages represented by symbols from finite alphabets between the variable nodes (VNs) and the check nodes (CNs). Moreover, they utilize lookup tables (LUTs) with single input to carry out the node updates. These LUTs are carefully designed based on the density evolution (DE) [4] with a selected coarse quantization scheme, which aims to maximize the mutual information (MI) between the coded bits and the exchanged messages within the decoders. We hereby call this type of FAIDs the mutual-information-maximizing FAIDs (MIM-FAIDs). More specifically, the MIM-FAIDs [8–10,12,14,15] implement the

coarse quantization scheme by dynamic programming (DP) [21], which has been proved to be optimal with respect to maximizing MI. In [11,13], the LUTs of the MIM-FAIDs are designed based on the information bottleneck (IB) method, which makes use of machine learning rather than DP in the design process. Furthermore, the MIM-FAID proposed in [16] constructs the LUTs by using a hierarchical dynamic quantization which is a greedy quantization scheme similar to the IB method and requires less computational complexity compared to DP. In addition, there are two different node updating architectures considered by the above MIM-FAIDs. One is designing multiple sets of concatenated two-input LUTs for decoding, e.g., [8,11,12], where each set of LUTs is dedicated to updating the nodes of a specific degree at each iteration. However, the concatenated LUTs only focus on maximizing MI between two consecutive tables after quantization and hence lead to a loss of MI, which may deteriorate the decoder performance. Moreover, the memory requirement for storing the LUTs may be intolerable because the number of LUTs increases significantly when the node degree or the decoding iteration becomes large. To reduce the memory demand and preserve more MI after quantization, the MIM-FAIDs were proposed in [9,10,14–16], which performed the node updates following a reconstruction–calculation–quantization architecture. Specifically, the FAID in [16] utilizes real additions and multiple sets of single-input LUTs with real-valued entries to update all nodes of different degrees. The mutual-information-maximizing (MIM) quantized decoders in [9,10,14,15] adopt integer additions and the LUTs of integer entries for practical consideration.

To accelerate the convergence speed, some MIM-FAIDs, i.e., [13,15,16], further consider either a layered schedule [22] or a shuffled schedule [23]. For example, the layered MIM-FAID [13] is designed by the IB method for decoding the regular LDPC codes. The LUTs of the FAID in [16] are constructed based on the layered schedule with a high-precision uniform channel quantizer. The MIM quantized shuffled min-sum (MIM-QSMS) decoder was proposed in [15], which designs the LUTs by considering the shuffled decoding schedule. All of these MIM-FAIDs with different decoding schedules, e.g., [15,16], are designed for a particular LDPC code with a fixed code rate, which cannot be used to decode LDPC codes with different code rates. However, the rate-compatible quasi-cyclic LDPC (RC-QC-LDPC) codes are preferred in many practical applications such as data storage systems [24,25]. An LDPC decoder that fails to support rate compatibility may incur a high complexity for hardware implementations. Although the rate-compatible MIM-FAIDs were investigated in [11,14], they were only designed based on the flooding schedule [26]. Therefore, how to design the MIM-FAID with a layered/shuffled schedule for LDPC codes with different code rates is still a challenging problem.

In this paper, we develop an MIM-QSMS decoder for decoding RC-QC-LDPC codes, which is referred to as the rate-compatible MIM-QSMS (RC-MIM-QSMS) decoder. Compared to other MIM-FAIDs in the literature, our proposed RC-MIM-QSMS decoder integrates the shuffled decoding schedule and the rate compatibility in a single round of the LUT design process. To the best of our knowledge, this is the first design for the FAID to support the rate compatibility with a shuffled schedule. In particular, we modify the DE in [15] and propose the shuffled MIM-DE (SMIM-DE) by considering the weighted expectation of the probability mass functions (pmfs) and the joint degree distributions. Based on the SMIM-DE, we are able to construct LUTs that vary with different layers and iterations for decoding RC-QC-LDPC codes. Moreover, an LUT optimization method is further proposed to generate a unique set of LUTs that only vary with decoding iterations. In this way, the memory requirement for storing the LUTs can be significantly reduced. We conduct a comprehensive evaluation on the proposed RC-MIM-QSMS decoder in terms of the error rate performance, the convergence speed, and the memory requirement for decoding. We demonstrate that the proposed RC-MIM-QSMS decoder surpasses the floating-point shuffled belief propagation (SBP) decoder [23] in the high signal-to-noise (SNR) region and has comparable convergence speed to other state-of-the-art MIM-FAIDs. More importantly, the RC-MIM-QSMS decoder can save up to 85.43% memory requirement compared to the benchmark MIM-FAIDs.

The rest of this paper is organized as follows. Section 2 provides the preliminaries of this work including the notations, the QC-LDPC codes, the shuffled min-sum (SMS) decoder [27], and the decoding framework of the MIM-QSMS decoder [15]. The proposed SMIM-DE and the LUT optimization method for designing the RC-MIM-QSMS decoder are illustrated in Section 4. In Section 5, we evaluate the proposed RC-MIM-QSMS decoder from the aspects of the error rate performance, the convergence speed, and the memory usage for implementing decoding. Section 6 concludes this paper.

## 2. Preliminaries

### 2.1. Notations

In this paper, calligraphy capitals denote alphabet sets. Normal capitals denote the random variables. Lower-case letters denote the realization of a random variable. Boldface letters are used to define a vector or matrix.

### 2.2. QC-LDPC Codes

QC-LDPC codes belong to a class of structured LDPC codes which can be represented by an $M_b \times N_b$ base matrix $\mathbf{H}_b$. Each element in $\mathbf{H}_b$ corresponds to a circulant permutation matrix [28] of size $Z \times Z$ such that we use integers between 0 and $Z - 1$ to specify the position of the one-entry in the first row of the circulant and use $\infty$ to represent an all-zero matrix. Let $\mathbf{H}$ be the parity-check matrix (PCM) of a QC-LDPC code. Accordingly, there are $M = M_b \cdot Z$ rows and $N = N_b \cdot Z$ columns in $\mathbf{H}$.

### 2.3. The SMS Decoder

Consider the Tanner graph [29] representation of the PCM $\mathbf{H}$ for a QC-LDPC code. For any positive integer $\omega$, we define the set $[\omega] = \{0, 1, \ldots, \omega - 1\}$. For $n \in [N]$ and $m \in [M]$, we denote the $n$th variable node (VN) and the $m$th check node (CN) in the Tanner graph by $v_n$ and $c_m$, respectively. Note that $v_n$ and $c_m$ are known as neighbors and connected to each other if there is a one-entry in the $m$th row and $n$th column of $\mathbf{H}$. Define the index sets of neighboring nodes of $v_n$ and $c_m$ by $\mathcal{N}(v_n)$ and $\mathcal{N}(c_m)$, respectively. For a set $\mathcal{A}$ and an element $a \in \mathcal{A}$, we denote the set with index $a$ excluded by $\mathcal{A} \setminus a$.

The shuffled decoding schedule was first proposed in [23] for the belief propagation (BP) algorithm to reduce the loading latency of the exchanged messages. For practical concerns, we focus on the low-complexity SMS decoder [27] and briefly introduce the details as follows. Denote the check-to-variable (C2V) message sent from $c_m$ to $v_n$ at the $t$th iteration by $S_{mn}^{(t)}$, and denote the variable-to-check (V2C) message sent from $v_n$ to $c_m$ by $R_{nm}^{(t)}$. Here, $t = 1, 2, \ldots, T_{\max}$, where $T_{\max}$ is a maximum preset number of iterations. Define $L(v_n)$ as the channel output of the node $v_n$. At each iteration, the SMS decoder updates the exchanged (C2V/V2C) messages by

$$S_{mn}^{(t)} = \alpha \cdot \prod_{n' \in \mathcal{N}(c_m) \setminus n} \mathrm{sgn}(R_{n'm}^{(t - \mathbb{I}(n' > n))}) \cdot \min_{n' \in \mathcal{N}(c_m) \setminus n} \left| R_{n'm}^{(t - \mathbb{I}(n' > n))} \right|, \tag{1}$$

$$R_{nm}^{(t)} = L(v_n) + \sum_{m' \in \mathcal{N}(v_n) \setminus m} S_{m'n}^{(t)}, \tag{2}$$

where $\alpha$ is the normalized factor and $\mathbb{I}$ is the indicator function operating as

$$\mathbb{I}(statement) = \begin{cases} 1, & statement \text{ is true}, \\ 0, & \text{otherwise}. \end{cases} \tag{3}$$

Note that for $t = 0$, we have $R_{nm}^{(0)} = L(v_n)$. Denote a posterior message of the node $v_n$ by $Q_n$, which can be computed at the end of each iteration by

$$Q_n = R_{nm}^{(t)} + S_{mn}^{(t)}. \tag{4}$$

In practice, the shuffled decoding schedule is generally conducted by partitioning the PCM **H** into $N_b$ groups, so-called the layers, where each layer consists of $Z$ VNs and corresponds to a QC column block of **H**. Note that we consider that any two columns in each layer have at most a one-entry in the same row. Therefore, the layer $j$ of **H** is equivalent to the $j$th column of the base matrix $\mathbf{H}_b$. Within a layer, the SMS decoder updates the exchanged messages in parallel and the decoding process proceeds in serial among consecutive layers.

## 3. The Related Work: Decoding Framework of an MIM-QSMS Decoder

The MIM-QSMS decoder [15] is one type of FAIDs that exchange the messages represented by symbols from finite alphabets within the decoder. The precision of the MIM-QSMS decoder is predetermined according to system constraints, which can be represented by the tuple $(q_m, q_v)$. More specifically, $q_m$ is the bit width of the exchanged messages and $q_v$ refers to the bit width of the a posteriori message. Figure 1 demonstrates the decoding framework of a $(q_m, q_v)$ MIM-QSMS decoder with respect to the node $v_n$ of degree $d_n$ at the $t$th iteration. Suppose that the MIM-QSMS decoder proceeds based on the base matrix $\mathbf{H}_b$ as described in Section 2.3. Therefore, we have $j = \lfloor n/Z \rfloor$ and $m_k \in \mathcal{N}(v_n) = \{m_1, m_2, \dots, m_{d_n}\}$.

- CN update: The MIM-QSMS decoder adopts the min operation [15] for the CN update, where we denote the CN update function by $\Phi_c^{\mathrm{MS}}$. Assume that $R_{n'm_k} \in \mathcal{R} = [2^{q_m}]$ for all $n' \in \mathcal{N}(c_{m_k}) \backslash n$ are the V2C message symbols received at a neighboring node $c_{m_k}$ connecting to the node $v_n$. Let $f(\cdot)$ be a function mapping the V2C message symbols $0, 1, \dots, 2^{q_m} - 1$ to integers $2^{q_m-1}, \dots, 1, -1, \dots -2^{q_m-1}$, respectively. At the $t$th iteration, the C2V message symbol $S_{m_k n}^{(t)} \in \mathcal{S} = [2^{q_m}]$ is computed by

$$
S_{m_k n}^{(t)} = \Phi_c^{\mathrm{MS}}\left( \left\{ R_{n'm_k}^{(t - \mathbb{I}(\lfloor n'/Z \rfloor > j))} : n' \in \mathcal{N}(c_{m_k}) \backslash n \right\} \right)
$$

$$
= f^{-1}\left( \prod_{n' \in \mathcal{N}(c_{m_k})\backslash n} \mathrm{sgn}\left( f\left( R_{n'm_k}^{(t - \mathbb{I}(\lfloor n'/Z \rfloor > j))} \right) \right) \cdot \min_{n' \in \mathcal{N}(c_{m_k})\backslash n} \left( \left| f\left( R_{n'm_k}^{(t - \mathbb{I}(\lfloor n'/Z \rfloor > j))} \right) \right| \right) \right), \quad (5)
$$

  where $f^{-1}(\cdot)$ is the inverse function of $f(\cdot)$.
- VN update: The VN update of the MIM-QSMS decoder computes the V2C messages $R_{nm_k} \in \mathcal{R} = [2^{q_m}]$ for all $m_k \in \mathcal{N}(v_n)$, which contains three steps, i.e., reconstruction, calculation, and quantization. We denote the quantized channel output of the node $v_n$ by $L_n \in \mathcal{L} = [2^{q_m}]$. Denote the C2V message coming from the node $c_{m_k}$ to the node $v_n$ by $S_{m_{k'}}n \in \mathcal{S} = [2^{q_m}], m_{k'} \in \mathcal{N}(v_n) \backslash m_k$. During the VN update, the quantized channel output $L_n$ and all C2V messages $S_{m_{k'}}n$ are firstly mapped to the computational messages based on reconstruction LUTs $\phi_{ch}$ and $\phi_v$, respectively. The computational messages are essentially the integers of bit width much larger than $q_m$. Following that, the VN update function $\Phi_v$ is adopted in the calculation step to compute the V2C computational messages, denoted by $B_{nm_k} \in \mathcal{B} = [2^{q_v}]$, as

$$
B_{nm_k}^{(t)} = \Phi_v\left( \left\{ L_n, S_{m_{k'}n}^{(t)} : \forall m_{k'} \in \mathcal{N}(v_n) \backslash m_k \right\} \right)
$$

$$
= \phi_{ch}(L_n) + \sum_{m_{k'} \in N(v_n) \backslash m_k} \phi_v(S_{m_{k'}n}^{(t)}). \quad (6)
$$

Finally, each V2C computational message is quantized into a V2C message in $\mathcal{R}$ by a quantization LUT, denoted by $\Gamma_v$. Note that the reconstruction LUTs and quantization LUTs of the MIM-QSMS decoder in [15] originally vary with decoding iterations and layers. Here, we omitted the associated iteration and layer of the LUTs in (6) for simplicity.

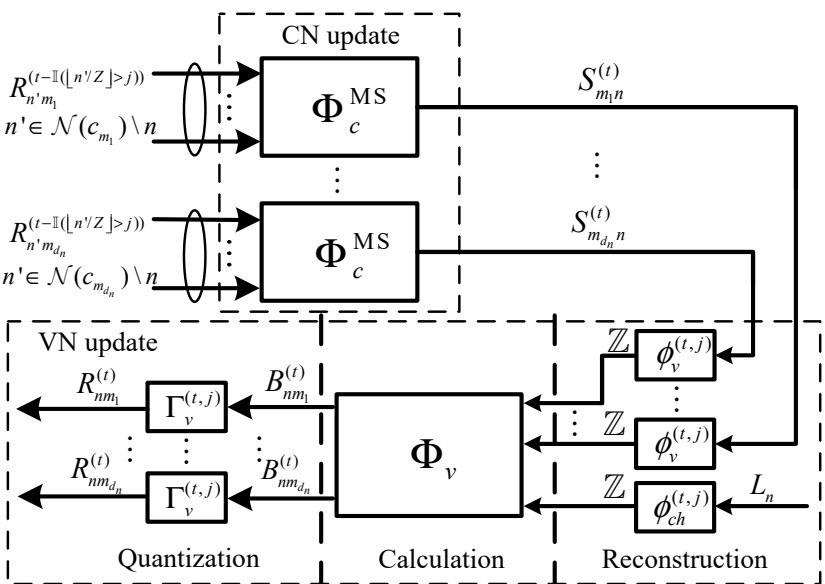

**Figure 1.** The decoding architecture of the layer-specific MIM-QSMS decoder [15].

Based on the decoding architecture in Figure 1, the main idea of designing the MIM-QSMS decoder in [15] is to construct the LUTs by tracking the probability mass functions (pmfs) of exchanged messages via DE [4]. To be more specific, the reconstruction LUTs are generated by scaling the LLR values associated with each message symbol in $\mathcal{L}$ and $\mathcal{S}$, respectively. The quantization LUTs are designed to be an optimal sequential deterministic quantizer (SDQ) by using dynamic programming (DP) [21].

## 4. Design of MIM-QSMS Decoder for RC-QC-LDPC Codes

The MIM-QSMS decoder in [15] is designed for an LDPC code with a fixed code rate, where we refer to this type of decoder as the rate-dependent MIM-QSMS (RD-MIM-QSMS) decoder. Obviously, decoding LDPC codes with various code rates using the RD-MIM-QSMS decoder requires different sets of LUTs corresponding to each code rate. Thus, it not only requires multiple rounds of design process to construct the LUTs but also a large amount of memory usage for hardware implementation. Although constructing LUTs used for decoding RC-QC-LDPC codes has been proposed in [11,14], these methods only focus on the flooding schedule and cannot be directly applied to a shuffled decoder. This is because the updating order of the exchanged messages in a shuffled decoder is different from that in a decoder with a flooding schedule. Motivated by the above observations, in this section, we propose a design method of the RC-MIM-QSMS decoder, which is capable of decoding RC-QC-LDPC codes based on only one set of LUTs with much less memory requirement.

### 4.1. Proposed SMIM-DE

In [14], the joint edge-degree distributions were considered by the DE in the LUT design process to support rate compatibility. Inspired by this, we propose a SMIM-DE to construct the LUTs of the RC-MIM-QSMS decoder, which is modified from the DE in [14] and considers both the joint edge-degree distributions and the shuffled decoding schedule. Let $\mathcal{D}_c$ and $\mathcal{D}_v$ be the sets of CN and VN degrees for $K$ target LDPC codes, respectively. Then,

$$
\begin{aligned}
\mathcal{D}_c &= \bigcup_{k=1}^{K} \mathcal{D}_{c,k} = \{d_{c,1}, d_{c,2}, \ldots, d_{c,\max}\}, \\
\mathcal{D}_v &= \bigcup_{k=1}^{K} \mathcal{D}_{v,k} = \{d_{v,1}, d_{v,2}, \ldots, d_{v,\max}\},
\end{aligned}
\tag{7}
$$

where $\mathcal{D}_{c,k}$ and $\mathcal{D}_{v,k}$ represent the sets of the CN and VN degrees of the $k$th target LDPC code, respectively. The joint edge-degree distributions can be given by [14]

$$\rho(\xi) = \sum_{d \in \mathcal{D}_c} \rho_d \xi^{d-1}, \; \theta(\xi) = \sum_{d \in \mathcal{D}_v} \theta_d \xi^{d-1}, \tag{8}$$

where $\rho_d$ and $\theta_d$ are the fractions of edges connected to the CNs of degree $d$ and the VNs of degree $d$, respectively, in the Tanner graphs of all $K$ target LDPC codes. Similar to the MIM-QSMS decoder described in Section 3, we consider the LUT design to be processed on the base matrices of the target LDPC codes. For the $k$th target LDPC code, we assume its base matrix has $N_{b,k}$ columns, which correspond to $N_{b,k}$ layers for the shuffled schedule. Define $N_{b,\max} = \max\{N_{b,k} : k = 1, 2, \ldots, K\}$ as the maximum number of layers for $K$ target LDPC codes. Let $X \in \mathcal{X} = \{0, 1\}$ be the random variable for the coded bit. For a random variable $A$ taking values from $\mathcal{A}$, we denote $P_{A|X}(a|x)$ as the pmf of $A = a \in \mathcal{A}$ conditioned on $X = x \in \mathcal{X}$. We also define the random variable for the channel output by $L \in \mathcal{L}$, and the random variable for the V2C (respectively, C2V) message by $R \in \mathcal{R}$ (respectively, $S \in \mathcal{S}$). For convenience, we use the superscript $(t, j)$ to represent a pmf computed at the $j$th layer and $t$th iteration, where $j = 1, 2, \ldots, N_{b,\max}$ and $t = 1, 2, \ldots, T_{\max}$. The SMIM-DE is illustrated as follows.

### 4.1.1. Channel Quantization

As shown in the literature, e.g., [9,11,12,16], the design of the FAIDs is conducted under a discrete additive white Gaussian noise (AWGN) channel with a properly selected noise standard deviation. Therefore, we first present the channel quantization considered by the SMIM-DE, which is used to discretize a continuous AWGN channel at the beginning of the design process. Assume that $X$ is modulated by binary phase-shift keying (BPSK). Given a preset bit width $q_m$, we first uniformly quantize the AWGN channel into a discrete memoryless channel with $Y$ outputs, where $Y \gg 2^{q_m}$, and we set $Y = 2002$ in this paper. Then, we adopt DP to find an optimal SDQ [21] for a $q_m$-bit channel output, which aims to maximize the mutual information between $X$ and $L$. In this way, we can obtain the conditional pmf $P_{L|X}$ and the corresponding quantization LUT of the SDQ, denoted by $\Gamma_{ch}$. The constructed quantization LUT is essentially a threshold set $\Gamma_{ch} = \{\gamma_k : k = 1, 2, \ldots, 2^{q_m} - 1\}$ with $\gamma_1 > \gamma_2 > \cdots > \gamma_{2^{q_m}-1}$, which operates as

$$\Gamma_{ch}(x) = \begin{cases} 0, & x \geq \gamma_1, \\ 2^{q_m} - 1, & x < \gamma_{2^{q_m}-1}, \\ k, & \gamma_k > x \geq \gamma_{k+1}. \end{cases} \tag{9}$$

Based on $P_{L|X}$ and $\Gamma_{ch}$, we next discuss the node updates of the SMIM-DE.

### 4.1.2. CN Update

In [15], we assume that the V2C messages sent to the CNs located in the same layer have equal contributions so that we consider their expected pmfs for the CN update. However, this is not the case in designing the RC-MIM-QSMS decoder because the CNs at the same layer are supposed to receive the V2C messages not only from different layers but also from different target LDPC codes when using the joint edge-degree distributions. Since the number of layers may be different for each target LDPC code, we consider the weighted expectation of the V2C message pmfs in the proposed SMIM-DE by introducing a parameter called the weight coefficient. Define $\mathbf{I}_k$ of size $N_{b,\max}$ as the layered indicator of the $k$th target LDPC code. Each entry $i_{k,j} \in \mathbf{I}_k$ for $j = 1, 2, \ldots, N_{b,\max}$ is of binary values such that

$$i_{k,j} = \begin{cases} 1, & j \leq N_{b,k}, \\ 0, & \text{otherwise}, \end{cases} \tag{10}$$

where the value 1 means that the corresponding layer exists for the $k$th target LDPC code. At the $j$th layer, we define the weight coefficient of the V2C message pmfs as

$$w_j = \frac{\sum_{k=1}^{K} i_{k,j}}{\sum_{k=1}^{K} \sum_{j=1}^{N_{b,\max}} i_{k,j}}. \tag{11}$$

At the $t$th iteration, a CN of degree $d$ ($d \in \mathcal{D}_c$) in layer $j$ receives the V2C messages that were updated at iteration $t-1$ from layers $j' > j$ and have been updated at iteration $t$ from layers $j' < j$. Therefore, the expected pmf of the V2C message received at layer $j$ can be expressed as

$$\tilde{P}_{R|X}^{(t,j)}(r|x) = \frac{\sum_{j' \neq j,\, j'=1}^{N_{b,\max}} w_{j'} \cdot P_{R|X}^{(t-\mathbb{I}(j'>j),\, j')}(r|x)}{\sum_{j' \neq j,\, j'=1}^{N_{b,\max}} w_{j'}}. \tag{12}$$

Note that we have $P_{R|X}^{(0,j)} = P_{L|X}$ for $j = 1, 2, \ldots, N_{b,\max}$. Define the vector of V2C messages received at a degree-$d$ ($d \in \mathcal{D}_c$) CN by $\mathbf{R} \in \mathcal{R}^{d-1}$, where $\mathbf{r} = (r_1, r_2, \ldots, r_{d-1}) \in \mathcal{R}^{d-1}$ is a realization of $\mathbf{R}$. Denote $\mathbf{x} = (x_1, x_2, \ldots, x_{d-1})$ as the vector of coded bits associated to its neighboring VNs. With the independent and identical distribution (i.i.d.) assumption [4], the joint pmf at layer $j$ and iteration $t$ is given by

$$\tilde{P}_{\mathbf{R}|X}^{(t,j)}(\mathbf{r}|x) = \left(\frac{1}{2}\right)^{d-2} \sum_{\mathbf{x}:\oplus\mathbf{x}=x} \prod_{k=1}^{d-1} \tilde{P}_{R|X}^{(t,j)}(r_k|x_k), \tag{13}$$

where $x \in \mathcal{X}$ is a realization of $X$, and $\oplus\mathbf{x} = x$ means the checksum of the CN is satisfied. For each received realization $\mathbf{r}$, the CN update function $\Phi_c^{\mathrm{MS}}$ computes the corresponding output $s$ as in (5). By considering the fraction $\rho_d$ in (8), the conditional pmf of the C2V message at layer $j$ and iteration $t$ can be obtained by

$$P_{S|X}^{(t,j)}(s|x) = \sum_{d \in D_c} \rho_d \cdot \sum_{\substack{\mathbf{r} \in \mathcal{R}^{d-1}, \\ \Phi_c^{\mathrm{MS}}(\mathbf{r})=s}} \tilde{P}_{\mathbf{R}|X}^{(t,j)}(\mathbf{r}|x). \tag{14}$$

Note that the computation of (14) can be conducted recursively such as in [4,16] to achieve low complexity, which is not a big issue for an off-line process.

### 4.1.3. VN Update

For the VN update, the SMIM-DE calculates the conditional pmf $P_{R|X}^{(t,j)}$ for layer $j$ at the $t$th iteration following the reconstruction, calculation, and quantization steps. Based on the conditional pmfs $P_{S|X}^{(t,j)}$ and $P_{L|X}$, we denote $h(l) = \log(P_{L|X}(l|0)/P_{L|X}(l|1))$ for $l \in \mathcal{L}$, and $h^{(t,j)}(s) = \log(P_{S|X}^{(t,j)}(s|0)/P_{S|X}^{(t,j)}(s|1))$ for $s \in \mathcal{S}$, respectively. Inspired by [15], the reconstruction LUTs associated to a degree-$d$ ($d \in \mathcal{D}_v$) VN at layer $j$ and iteration $t$ can be generated by

$$\begin{cases} \phi_{ch}^{(t,j)}(l) = \mathrm{round}(\eta \cdot |h(l)|), \\ \phi_{v}^{(t,j)}(s) = \mathrm{round}(\eta \cdot |h^{(t,j)}(s)|), \end{cases} \tag{15}$$

where $\mathrm{round}(x)$ returns the closest integer to $x$, and $\eta$ is the scaling factor, i.e.,

$$\eta = \frac{2^{q_v-1}-1}{d+1} \cdot \max\Big(\{|h^{(t,j)}(s)| : s \in \mathcal{S}\} \cup \{|h(l)| : l \in \mathcal{L}\}\Big), \tag{16}$$

to allow the maximum number of $q_v$ bit width for the VN update. Assume that the C2V messages sent to the VNs in the same layer are i.i.d. We denote the vector of the C2V messages and the channel output received at a degree-$d$ VN by $(L, \mathbf{S}) \in \mathcal{L} \times \mathcal{S}^{d-1}$, where

$\mathbf{s} = (s_1, s_2, \ldots, s_{d-1}) \in \mathcal{S}^{d-1}$ and $l \in \mathcal{L}$ is a realization of $\mathbf{S}$ and $L$, respectively. The joint pmf at layer $j$ and iteration $t$ can be computed by

$$P^{(t,j)}_{L,\mathbf{S}|X}(l, \mathbf{s}|x) = P_{L|X}(l|x) \prod_{k=1}^{d-1} P^{(t,j)}_{S|X}(s_k|x). \tag{17}$$

Define $\mathcal{B}^{(t,j)}$ as the alphabet set of the V2C computational message calculated at layer $j$ and iteration $t$ for the VN update, where $B \in \mathcal{B}$ is the random variable for the V2C computational message. After reconstruction, the VN update function $\Phi_v$ computes the realization $b$ of $B$ based on each input $(l, \mathbf{s}) \in \mathcal{L} \times \mathcal{S}^{d-1}$ to form the alphabet set $\mathcal{B}^{(t,j)}$. According to the fraction $\theta_d$ given by (8), the conditional pmf of the V2C computational message is represented by

$$P^{(t,j)}_{B|X}(b|x) = \sum_{d \in \mathcal{D}_v} \theta_d \cdot \sum_{\substack{(l,\mathbf{s}) \in \mathcal{L} \times \mathcal{S}^{d-1}, \\ \Phi_v(l,\mathbf{s})=b}} P^{(t,j)}_{L,\mathbf{S}|X}(l, \mathbf{s}|x). \tag{18}$$

Similar to (14), we can compute (18) in a recursive manner for a low computational complexity. With $P^{(t,j)}_{B|X}$ and $\mathcal{B}^{(t,j)}$, we perform DP to design the quantization LUT $\Gamma^{(t,j)}_v$ as an optimal SDQ [21] and obtain the conditional pmf $P^{(t,j)}_{R|X}$ such that

$$[P^{(t,j)}_{R|X}, \Gamma^{(t,j)}_v] = \mathbf{DP}(\mathcal{B}^{(t,j)}, P^{(t,j)}_{B|X}), \tag{19}$$

where $\Gamma^{(t,j)}_v$ operates the same as $\Gamma_{ch}$ in (9).

### 4.2. LUT Optimization

The SMIM-DE described above tracks the evolution of $P^{(t,j)}_{R|X}$ and $P^{(t,j)}_{S|X}$ at each layer and each iteration to construct the LUTs varying with different layers and iterations for decoding. For a predetermined maximum number of decoding iterations $T_{\max}$ and $N_{b,\max}$ layers of the target LDPC codes, the memory requirement for storing the LUTs constructed in this way grows linearly with $N_{b,\max} \times T_{\max}$, which increases significantly in particular for the target LDPC codes with large $N_{b,\max}$. Thus, it is necessary to optimize the SMIM-DE to generate LUTs with less memory demand for hardware implementation. To this end, we propose an optimization method for the RC-MIM-QSMS decoder in a postprocessing manner to design the LUTs that only vary with decoding iterations, which we refer to as the iteration-specific LUTs in the rest of this paper. In particular, we first conduct the SMIM-DE at each iteration to obtain the conditional pmfs $P^{(t,j)}_{S|X}$ for all layers. Similar to the case of the CN update, we assume that the C2V messages sent to a VN at the $t$th iteration are from $N_{b,\max}$ layers of all target LDPC codes. Thus, we combine the conditional pmfs from $N_{b,\max}$ layers into the iteration-specific pmf by considering the weighted expectation of the C2V message pmfs at the $t$th iteration such that

$$\tilde{P}^{(t)}_{S|X}(s|x) = \frac{\sum_{j=1}^{N_{b,\max}} w_j \cdot P^{(t,j)}_{S|X}(s|x)}{\sum_{j=1}^{N_{b,\max}} w_j}. \tag{20}$$

The iteration-specific reconstruction LUTs $\phi^{(t)}_{ch}$ and $\phi^{(t)}_v$ can be constructed by replacing $P^{(t,j)}_{S|X}$ in (15) and (16) with $\tilde{P}^{(t)}_{S|X}$. The quantization LUT $\Gamma^{(t)}_v$ for each iteration can also be obtained following (17)–(19). With these iteration-specific LUTs, the conditional pmf $P^{(t,j)}_{R|X}$ requires to be further updated by the SMIM-DE for designing the LUTs at the next iteration. The LUT optimization repeats at each iteration until the preset maximum number of iterations $T_{\max}$ is reached.

To summarize, we present the design flow of the RC-MIM-QSMS decoder with the LUT optimization in Algorithm 1. Note that by adopting the LUT optimization, the RC-MIM-QSMS decoder requires only four LUTs per iteration, which consists of two reconstruction LUTs ($\phi_{ch}^{(t)}$ and $\phi_v^{(t)}$) and two quantization LUTs ($\Gamma_{ch}$ and $\Gamma_v^{(t)}$). For the decoding process, all the LUTs are fixed for different received signal-to-noise ratios (SNRs). Furthermore, $\Gamma_{ch}$ is fixed for all iterations while the other LUTs may vary with different iterations.

---

**Algorithm 1** The Design Flow of MIM-QSMS Decoder with LUT Optimization

---

**Input:** $q_m$, $q_v$, $\rho(\xi)$, $\theta(\xi)$, $\sigma_d$, $N_{b,\max}$, $T_{\max}$.
**Output:** $\Gamma_{ch}$, $\Gamma_v^{(t)}$, $\phi_{ch}^{(t)}$, $\phi_v^{(t)}$.
1: Compute $P_{L|X}$ and $\Gamma_{ch}$ as described in Section 4.1.1
2: Compute the weight coefficient $w_j$ using (11) for $j = 1, 2, \ldots, N_{b,\max}$
3: $P_{R|X}^{(0,j)} = P_{L|X}$, $j = 1, 2, \ldots, N_{b,\max}$
4: **for** $t = 1 : T_{\max}$ **do**
5:     **for** $j = 1 : N_{b,\max}$ **do**
6:         Compute $\tilde{P}_{R|X}^{(t,j)}$ and $P_{S|X}^{(t,j)}$ based on (12) and (14), respectively
7:         Construct $\phi_{ch}^{(t,j)}$ and $\phi_v^{(t,j)}$ based on (15)
8:         Compute $\mathcal{B}^{(t,j)}$ and $P_{B|X}^{(t,j)}$ according to (6) and (18), respectively
9:         Perform $[P_{R|X}^{(t,j)}, \Gamma_v^{(t,j)}] = \mathbf{DP}(\mathcal{B}^{(t,j)}, P_{B|X}^{(t,j)})$
10:     **end for**
11:     Compute $\tilde{P}_{S|X}^{(t)}$ by (20)
12:     Construct $\phi_{ch}^{(t)}$ and $\phi_v^{(t)}$ with $\tilde{P}_{S|X}^{(t)}$ by (15)
13:     Determine $\Gamma_v^{(t)}$ with $\tilde{P}_{S|X}^{(t)}$ based on (19)
14:     **for** $j = 1 : N_{b,\max}$ **do**
15:         Perform **Steps** 6–8 using $\phi_{ch}^{(t)}$ and $\phi_v^{(t)}$
16:         Update $P_{R|X}^{(t,j)}$ using $\Gamma_v^{(t)}$ based on $\mathcal{B}^{(t,j)}$, $P_{B|X}^{(t,j)}$
17:     **end for**
18: **end for**

---

### 4.3. Remarks

Denote the noise standard deviation of the selected AWGN channel to design the RC-MIM-QSMS decoder as $\sigma_d$. Given the tuple $(q_m, q_v)$, the degree distributions $(\rho(\xi), \theta(\xi))$, and the maximum number of iterations $T_{\max}$, different choices of $\sigma_d$ result in different sets of reconstruction and quantization LUTs. Compared to the conventional DE [4] that focuses on the decoding error probability, the design of the RC-MIM-QSMS decoder is similar to that of other MIM-FAIDs, e.g., [12,15,16], which aim to achieve a certain MI value approaching 1 for the a posteriori messages. Let $\sigma^*$ be the optimal noise standard deviation for designing the RC-MIM-QSMS decoder, which is determined by following the steps below. Denote the alphabet set of a posteriori messages by $\mathcal{Q}$, where $Q \in \mathcal{Q}$ is the random variable for a posterior message. With the iteration-specific reconstruction LUTs and the conditional pmf $\tilde{P}_{S|X}^{(t)}$ given by (20), we first obtain $\mathcal{Q}$ from (6) and compute the conditional pmf $P_{Q|X}$ based on (18) by considering $(l, \mathbf{s}) \in \mathcal{L} \times \mathcal{S}^d$. Define $\hat{X}$ as the hard decision of $Q$, which takes values from $\mathcal{X} = \{0, 1\}$. We select $\sigma^*$ equal to the maximum $\sigma_d$ that achieves a mutual information between $X$ and $\hat{X}$ greater than $1 - \epsilon$ after $T_{\max}$ iterations, i.e.,

$$\sigma^* = \sup \left\{ \sigma_d : I^{(\sigma_d, T_{\max})}(X; \hat{X}) > 1 - \epsilon \right\}. \tag{21}$$

As shown in the literature [12,15,16], we set $\epsilon = 10^{-3}$ in this paper for achieving a desirable error rate performance across a wide range of SNRs. We note that the proposed RC-MIM-QSMS decoder directly determines the coded bits at each iteration according to $\hat{X}$ rather than using the quantization LUT for bit decision as in [14,15]. This is more efficient for hardware implementation compared to the RD-MIM-QSMS decoder [15] and the MIM-FAID [14].

In addition, we also notice that it is possible for the RC-MIM-QSMS decoder to cause a performance degradation compared to the RD-MIM-QSMS decoders since the joint degree distributions considered in the LUT design are mismatched with the specific degree distributions of any target LDPC codes. However, the performance loss can be limited within a certain range by selecting appropriate target LDPC codes for rate-compatible design. Intuitively, for preset bit widths ($q_m$, $q_v$) and the maximum number of iterations $T_{\max}$, the target LDPC codes that have the design noise standard deviations close to each other are preferable to be optimized jointly. This is because the MI values of these target LDPC codes are likely to approach 1 after $T_{\max}$ iterations. To verify this hypothesis, we quantify $\sigma_d$ for a given code rate $R_c$ by the corresponding design SNR, which is denoted as

$$\tau \overset{\Delta}{=} -10\log_{10}(2R_c \cdot \sigma_d^2) \text{ [dB]}. \tag{22}$$

Based on extensive simulation results, we let any two target LDPC codes have different values of $\tau$ within 1.2 dB for achieving a desirable error rate performance. As we can see in the later simulations, this criterion leads to the RC-MIM-QSMS decoders operating on three different code rates for the length-1296 IEEE 802.11n LDPC codes [30] and the fifth-generation (5G) LDPC codes [31].

## 5. Simulation Results and Discussion

In this section, we evaluate the performance of the proposed RC-MIM-QSMS decoder with respect to the frame error rate, the convergence speed, and the memory requirement via Monte-Carlo simulations. Moreover, we also include the performance of the floating-point SBP decoder [23], the conventional QSMS decoder [27], the rate-compatible FAID (RC-FAID) decoder with flooding schedule [11], the rate-compatible MIM quantized min-sum (RC-MIM-QMS) decoder [14][R2], and the RD-MIM-QSMS decoder [15] for comparison. We denote the bit width settings of the exchanged messages ($q_m$) and the a posteriori message ($q_v$) of different LDPC decoders by ($q_m$, $q_v$), and the floating-point precision is represented by "$\infty$".

We adopted a BPSK modulation and assumed the LDPC codewords were transmitted over the AWGN channels. We considered two types of LDPC codes which have moderate and short block lengths, respectively. One was the length-1296 LDPC codes adopted in the IEEE 802.11n standard [30] with code rates 2/3, 3/4, and 5/6, respectively. Another was the 5G LDPC codes constructed from a base graph one with lifting size 26 with code rates 3/4, 5/6, and 8/9 after rate matching [31]. Tables 1 and 2 show the degree distributions of the simulated codes and the design noise standard deviation ($\sigma_d$) for the associated MIM-QSMS decoder, respectively. Note that we designed the RC-FAID decoder based on DP [21] rather than the information bottleneck method [11] because DP proved to be optimal for maximizing MI. At least 300 error frames were collected at each simulated SNR. In addition, we set $\alpha = 0.8$ for the conventional QSMS decoder and $T_{\max} = 15$ for all decoders.

**Table 1.** Degree Distributions of the IEEE 802.11n LDPC Codes and the 5G LDPC Codes.

| Code Types | $R_c$ | Degree Distributions $(\rho(\xi), \theta(\xi))$ |
|---|---|---|
| 802.11n | 2/3 | $\rho(\xi) = \xi^{10}$<br>$\theta(\xi) = 0.1591\xi + 0.4091\xi^2 + 0.1591\xi^6 + \xi 0.2727\xi^7$ |
| | 3/4 | $\rho(\xi) = 0.3182\xi^{13} + 0.6818\xi^{14}$<br>$\theta(\xi) = 0.1136\xi + 0.4091\xi^2 + 0.4773\xi^5$ |
| | 5/6 | $\rho(\xi) = 0.7412\xi^{20} + 0.2588\xi^{21}$<br>$\theta(\xi) = 0.0706\xi + 0.1765\xi^2 + 0.7529\xi^3$ |
| 5G | 3/4 | $\rho(\xi) = 0.0246\xi^2 + 0.0574\xi^6 + 0.0656\xi^7 + 0.1475\xi^8 + 0.0820\xi^9 + 0.6230\xi^{18}$<br>$\theta(\xi) = 0.0492 + 0.0328\xi + 0.1230\xi^2 + 0.1967\xi^3$<br>$+0.4508\xi^4 + 0.0656\xi^7 + 0.0820\xi^9$ |
| | 5/6 | $\rho(\xi) = 0.0313\xi^2 + 0.0833\xi^7 + 0.0938\xi^8 + 0.7917\xi^{18}$<br>$\theta(\xi) = 0.0313 + 0.0625\xi + 0.2813\xi^2 + 0.5000\xi^3 + 0.0521\xi^4 + 0.0729\xi^6$ |
| | 8/9 | $\rho(\xi) = 0.0380\xi^2 + 0.9620\xi^{18}$<br>$\theta(\xi) = 0.0127 + 0.0759\xi + 0.7975\xi^2 + 0.0506\xi^3 + 0.0633\xi^4$ |

**Table 2.** Design Noise Standard Deviations for the RC-MIM-QSMS and RD-MIM-QSMS Decoders.

| Code Types | $R_c$ | RD-MIM-QSMS | | RC-MIM-QSMS | |
|---|---|---|---|---|---|
| | | (3, 7) | (4, 8) | (3, 7) | (4, 8) |
| 802.11n | 2/3 | 0.6877 | 0.7002 | 0.6058 | 0.6219 |
| | 3/4 | 0.6148 | 0.6267 | | |
| | 5/6 | 0.5392 | 0.5493 | | |
| 5G | 3/4 | | 0.6705 | | 0.6050 |
| | 5/6 | - | 0.5988 | - | |
| | 8/9 | | 0.5433 | | |

## 5.1. FER Performance

Figure 2 shows the FER performance of different decoders for the length-1296 IEEE 802.11n LDPC codes with code rates 2/3, 3/4, and 5/6. We can see that the proposed RC-MIM-QSMS decoder achieves almost the same FER compared to its rate-dependent counterparts for the same bit width settings and the same LDPC codes. Moreover, the $(4, 8)$-RC-MIM-QSMS decoder can outperform the $(4, 8)$-QSMS decoder by at least 0.1 dB and approaches the performance of the $(\infty)$-SBP decoder within 0.15 dB. In addition, the $(4, 8)$-RC-MIM-QSMS decoder also surpasses both the RC-MIM-QMS decoder[R2] and the RC-FAID decoder with the same bit width settings at most 0.25 dB for all simulated codes. With the $(3, 7)$ bit width settings, the RC-MIM-QSMS decoder has an FER performance close to the $(4, 8)$-QSMS decoder for code rate 5/6 and even outperforms the $(4, 8)$-QSMS decoder for code rates 3/4 and 2/3. Compared to the $(3, 7)$-RC-FAID decoder, the $(3, 7)$-RC-MIM-QSMS decoder achieves a performance gain of at least 0.2 dB for all code rates.

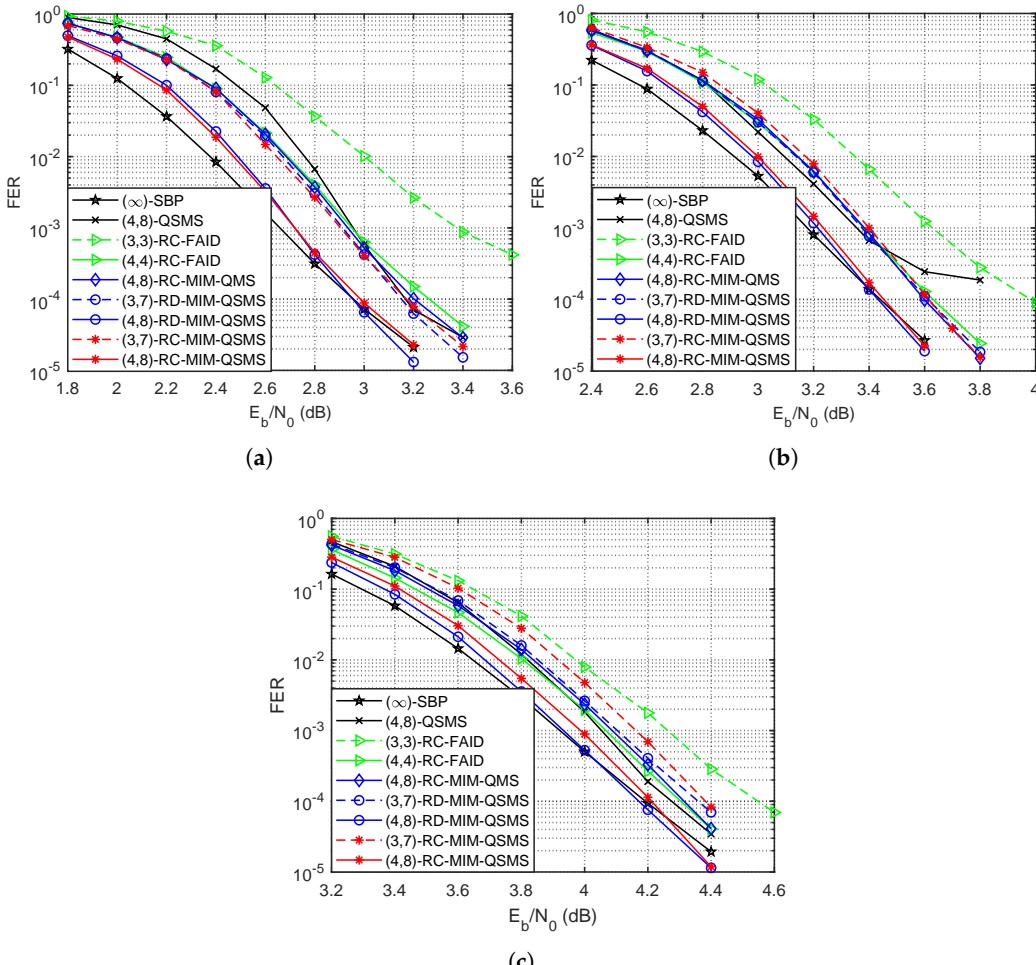

**Figure 2.** The FER performance of different decoders for the length-1296 IEEE 802.11n LDPC codes with code rates 2/3, 3/4, and 5/6. (**a**) $R_c = 2/3$. (**b**) $R_c = 3/4$. (**c**) $R_c = 5/6$.

Figure 3 depicts the FER performance of different decoders for the 5G LDPC codes with lifting size 26 and code rates 2/3, 3/4, and 5/6. As shown by the figure, the proposed $(4, 8)$-RC-MIM-QSMS decoder achieves almost the same FER compared to its rate-dependent counterparts for the same simulated code rates. Furthermore, the $(4, 8)$-RC-MIM-QSMS decoder performs better than the $(4, 8)$-QSMS decoder by up to 0.4 dB and approaches the performance of the $(\infty)$-SBP decoder within 0.2 dB. With the $(4, 8)$ bit width settings, the proposed RC-MIM-QSMS decoder also outperforms both the RC-MIM-QMS decoder[R2] and the RC-FAID decoder by at least 0.2 dB for all simulated code rates.

In the high SNR region, the proposed RC-MIM-QSMS decoder even shows slightly better error floor performance compared to the $(\infty)$-SBP decoder for both the 802.11n LDPC codes and the 5G LDPC code. This is because there are degree-two VNs in the Tanner graphs of the simulated codes, which results in trapping sets due to the cycles being confined among these degree-two VNs [32]. These trapping sets become the most harmful objects and cause error floor in the high SNR region for the BP decoder. Similar phenomena are also observed in the literature [8,10,14–16], which show that the MIM quantization schemes can assist to mitigate the negative impact of certain harmful objects in the code structure.

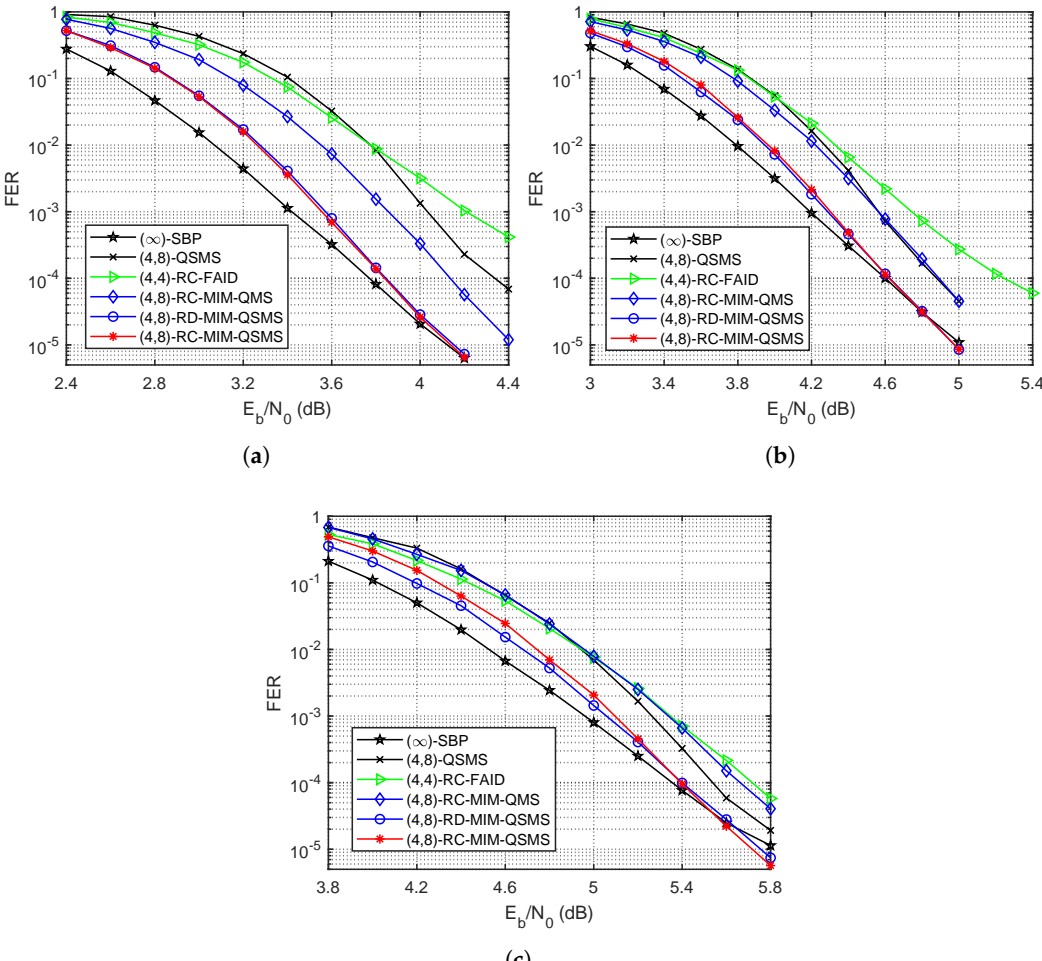

**Figure 3.** The FER performance of different decoders for the 5G LDPC codes with lifting size 26 and code rates 3/4, 5/6, and 8/9. (**a**) $R_c = 3/4$. (**b**) $R_c = 5/6$. (**c**) $R_c = 8/9$.

*5.2. Convergence Speed Analysis*

Apart from the FER performance, the convergence speed is another critical factor to assess the decoding latency. Define the average number of iterations required for decoding one codeword as $I_{\text{avg}}$. In Table 3, we compared the convergence speed of different quantized decoders with $q_m = 4$ for the 802.11n LDPC codes in perspective of $I_{\text{avg}}$. It can be seen that the proposed RC-MIM-QSMS decoder outperforms both the RC-FAID decoder and the RC-MIM-QMS decoder for all simulated code rates by reducing $I_{\text{avg}}$ up to 45.67%. The RC-MIM-QSMS decoder also achieves up to 34.07% less $I_{\text{avg}}$ than the conventional QSMS decoder in the low-to-moderate SNR region. We observe that $I_{\text{avg}}$ of the proposed RC-MIM-QSMS decoder is up to 7.67% less than the RD-MIM-QSMS decoder for the rate-2/3 802.11n LDPC code, and it slightly increases compared to the RD-MIM-QSMS decoder for the simulated LDPC codes with code rates 3/4 and 4/5. This is because the proposed RC-MIM-QSMS decoder is designed based on the joint degree distributions, which considers a larger portion of high-degree VNs with respect to the individual degree distributions of the rate-2/3 802.11n LDPC code. However, compared to the individual degree distributions of both rate-3/4 and rate-5/6 802.11n LDPC codes, there are a large portion of low-degree VNs considered by the joint degree distributions. The high-degree VNs lead to a faster convergence speed and the low-degree VNs have a slower convergence speed. Therefore, the proposed RC-MIM-QSMS decoder requires less $I_{\text{avg}}$ for lower code rates and more $I_{\text{avg}}$ for higher code rates compared to its rate-dependent counterparts.

**Table 3.** The average number of iterations $I_{avg}$ of different LDPC decoders with $q_m = 4$ for the 802.11n LDPC codes.

| | $E_b/N_0$ (dB) | 2 | 2.2 | 2.4 | 2.6 | 2.8 | 3 | 3.2 |
|---|---|---|---|---|---|---|---|---|
| | RC-FAID [11] | 13.01 | 11.49 | 9.82 | 8.22 | 7.05 | 6.15 | 5.45 |
| $I_{avg}$ | RC-MIM-QMS [14] | 13.02 | 11.35 | 9.75 | 8.12 | 6.94 | 6.06 | 5.38 |
| | Conventional QSMS [27] | 13.04 | 10.89 | 7.76 | 5.62 | 4.19 | 3.46 | 2.99 |
| | RD-MIM-QSMS [15] | 9.43 | 7.41 | 5.79 | 4.8 | 4.14 | 3.65 | 3.26 |
| $(R_c = 2/3)$ | RC-MIM-QSMS | 9.24 | 7.18 | 5.5 | 4.49 | 3.83 | 3.37 | 3.01 |
| | $E_b/N_0$ (dB) | 2.4 | 2.6 | 2.8 | 3 | 3.2 | 3.4 | 3.6 |
| | RC-FAID [11] | 13.03 | 11.02 | 9.07 | 7.41 | 6.19 | 5.28 | 4.61 |
| $I_{avg}$ | RC-MIM-QMS [14] | 13.17 | 11.11 | 9.14 | 7.47 | 6.23 | 5.32 | 4.64 |
| | Conventional QSMS [27] | 12.09 | 9.05 | 6.4 | 4.52 | 3.49 | 2.9 | 2.52 |
| | RD-MIM-QSMS [15] | 9.95 | 7.38 | 5.41 | 4.25 | 3.52 | 3.03 | 2.66 |
| $(R_c = 3/4)$ | RC-MIM-QSMS | 10.03 | 7.52 | 5.49 | 4.25 | 3.5 | 3.01 | 2.66 |
| | $E_b/N_0$ (dB) | 3.2 | 3.4 | 3.6 | 3.8 | 4 | 4.2 | 4.4 |
| | RC-FAID [11] | 10.71 | 8.3 | 6.46 | 5.09 | 4.2 | 3.58 | 3.11 |
| $I_{avg}$ | RC-MIM-QMS [14] | 11.28 | 8.79 | 6.85 | 5.39 | 4.43 | 3.76 | 3.25 |
| | Conventional QSMS [27] | 10.36 | 7.12 | 4.83 | 3.37 | 2.66 | 2.27 | 2.02 |
| | RD-MIM-QSMS [15] | 7.82 | 5.48 | 3.95 | 3.09 | 2.61 | 2.3 | 2.07 |
| $(R_c = 5/6)$ | RC-MIM-QSMS | 8.32 | 5.85 | 4.28 | 3.29 | 2.75 | 2.41 | 2.17 |

Table 4 demonstrates the $I_{avg}$ of different quantized decoders with $q_m = 4$ for the 5G LDPC codes. We can see that the proposed RC-MIM-QSMS decoder achieves less $I_{avg}$ for all simulated code rates by at least 37.71% in the moderate-to-high SNR region. Compared to the conventional QSMS decoder, the RC-MIM-QSMS decoder can reduce $I_{avg}$ by up to 42.04% for the rate-3/4 5G LDPC code. Similar to the case of the 802.11n LDPC codes, we also observe the phenomenon that $I_{avg}$ of the proposed RC-MIM-QSMS decoder has a minor reduction compared to the RD-MIM-QSMS decoder for the rate-3/4 5G LDPC code while it increases slightly for the code rates 5/6 and 8/9.

**Table 4.** The average number of iterations $I_{avg}$ of different LDPC decoders with $q_m = 4$ for the 5G LDPC codes.

| | $E_b/N_0$ (dB) | 2.8 | 3 | 3.2 | 3.4 | 3.6 | 3.8 | 4 |
|---|---|---|---|---|---|---|---|---|
| | RC-FAID [11] | 12.76 | 11.65 | 10.26 | 9.01 | 7.99 | 7.21 | 6.59 |
| $I_{avg}$ | RC-MIM-QMS [14] | 12.25 | 11.12 | 9.66 | 8.53 | 7.61 | 6.88 | 6.33 |
| | Conventional QSMS [27] | 12.33 | 10.37 | 8.07 | 6.2 | 4.66 | 3.77 | 3.16 |
| | RD-MIM-QSMS [15] | 8.16 | 6.48 | 5.34 | 4.53 | 3.96 | 3.54 | 3.21 |
| $(R_c = 3/4)$ | RC-MIM-QSMS | 7.63 | 6.01 | 4.9 | 4.16 | 3.64 | 3.27 | 2.97 |
| | $E_b/N_0$ (dB) | 3.6 | 3.8 | 4 | 4.2 | 4.4 | 4.6 | 4.8 |
| | RC-FAID [11] | 10.73 | 9.42 | 8.25 | 7.35 | 6.64 | 6.11 | 5.68 |
| $I_{avg}$ | RC-MIM-QMS [14] | 10.84 | 9.49 | 8.33 | 7.44 | 6.72 | 6.17 | 5.72 |
| | Conventional QSMS [27] | 8.34 | 6.35 | 4.87 | 3.77 | 3.13 | 2.71 | 2.44 |
| | RD-MIM-QSMS [15] | 5.84 | 4.76 | 4.03 | 3.53 | 3.18 | 2.91 | 2.69 |
| $(R_c = 5/6)$ | RC-MIM-QSMS | 6.27 | 5 | 4.21 | 3.63 | 3.22 | 2.91 | 2.67 |
| | $E_b/N_0$ (dB) | 4.4 | 4.6 | 4.8 | 5 | 5.2 | 5.4 | 5.6 |
| | RC-FAID [11] | 8.52 | 7.49 | 6.62 | 5.96 | 5.48 | 5.1 | 4.79 |
| $I_{avg}$ | RC-MIM-QMS [14] | 9.82 | 8.58 | 7.51 | 6.71 | 6.11 | 5.64 | 5.27 |
| | Conventional QSMS [27] | 6.28 | 4.61 | 3.64 | 2.95 | 2.57 | 2.33 | 2.17 |
| | RD-MIM-QSMS [15] | 4.52 | 3.71 | 3.2 | 2.84 | 2.59 | 2.4 | 2.26 |
| $(R_c = 8/9)$ | RC-MIM-QSMS | 5.81 | 4.7 | 3.96 | 3.44 | 3.07 | 2.79 | 2.58 |

### 5.3. Memory Requirement

We further investigated the overall memory requirement of the proposed RC-MIM-QSMS decoder and compare it to that of different quantized LDPC decoders. Here, we considered the decoders that are implemented based on software-defined radios or digital signal processors so that the LUTs are stored in memories [11]. We divided the memory into two types according to their usage, i.e., the memories for arithmetic calculation and those for storing the LUTs. For the conventional QSMS decoders, all $q_m$-bit V2C messages need to be stored for the arithmetic calculation of the node updates at each iteration [27], while the RC-FAID decoder with flooding schedule uses the memories for arithmetic calculation to store two $q_m$-bit C2V messages for each CN and one a posteriori message of $q_v$ bit width for each VN per iteration. Note that we considered the parity-check matrix of the rate-2/3 802.11n LDPC code to evaluate the maximum memory requirement for arithmetic calculation since it had the largest size among all simulated codes. Moreover, we assumed that the memories for arithmetic calculation could be reused between two consecutive iterations for improving efficiency. On the other hand, the memory requirement for storing one LUT could be computed by $(E \cdot q_t)/8$ in bytes, where $E$ is the number of entries in one LUT, and $q_t$ refers to the maximum bit width of an entry. According to [11], at each iteration, the RC-FAID decoder requires $d_{v,\max} - 1$ cascaded LUTs for updating the VNs of different degrees and one LUT for message alignment process. Since the RC-FAID decoder exchanges $q_m$-bit messages within the decoder and adopts two-input LUTs, we had $E = 2^{2q_m}$ for each LUT and $q_t = q_m$. In addition, there was one extra LUT of $q_t = 1$ at each iteration for making the hard decision. For the two MIM-QSMS decoders with $q_m = 4$, all LUTs had a single input so that there were two reconstruction LUTs of size $E = 2^{q_m}$ and one quantization LUT of size $E = 2^{q_m} - 1$. The bit width of each entry in each LUT could be obtained by $q_t = \lfloor \log_2(|z_{\max}|) \rfloor$, where $|z_{\max}|$ is the maximum magnitude of the entry.

Table 5 summarizes the overall memory requirements of different quantized LDPC decoders with $q_m = 4$ and $T_{\max} = 15$ for the 802.11n LDPC codes. As shown in the table, our proposed RC-MIM-QSMS decoder requires almost the same memory as the RC-MIM-QMS decoder and it only has a slight increase in the memory demand of 14.45% compared to that of the conventional QSMS decoder. More importantly, the RC-MIM-QSMS decoder can reduce the memory demand by 28.26% compared to its rate-dependent counterparts, and significantly saves 87.54% of memory demand when compared with the RC-FAID decoder.

**Table 5.** The Overall Memory Requirement for Different Quantized Decoders with $q_m = 4$ and $T_{\max} = 15$ for the 802.11n LDPC codes.

| Decoders | Arithmetic | LUTs $R_c = 2/3$ | $R_c = 3/4$ | $R_c = 5/6$ | Total |
|---|---|---|---|---|---|
| RC-FAID [11] | 0.42 kB | | 15.47 kB | | 15.89 kB |
| RC-MIM-QMS [14] | | | 0.26 kB | | 1.99 kB |
| Conventional QSMS [27] | 1.73 kB | | - | | 1.73 kB |
| RD-MIM-QSMS [15] | | 0.32 kB | 0.34 kB | 0.37 kB | 2.76 kB |
| RC-MIM-QSMS | | | 0.25 kB | | 1.98 kB |

Table 6 presents the overall memory requirements of different quantized LDPC decoders with $q_m = 4$ and $T_{\max} = 15$ for the 5G LDPC codes. It shows that our proposed RC-MIM-QSMS decoder only increases the memory demand by 16.81% compared to the conventional QSMS decoder and also requires less memory demand when compared with the RC-MIM-QMS decoder. More significantly, the RC-MIM-QSMS decoder requires

30.89% and 93.22% less memory demand compared to its rate-dependent counterparts and the RC-FAID decoder, respectively.

**Table 6.** The Overall Memory Requirement for Different Quantized Decoders with $q_m = 4$ and $T_{\max} = 15$ for the 5G LDPC codes.

| Decoders | Arithmetic | LUTs | | | Total |
| --- | --- | --- | --- | --- | --- |
| | | $R_c = 3/4$ | $R_c = 5/6$ | $R_c = 8/9$ | |
| RC-FAID [11] | 0.26 kB | | 19.22 kB | | 19.48 kB |
| RC-MIM-QMS [14] | | | 0.21 kB | | 1.34 kB |
| Conventional QSMS [27] | 1.13 kB | | - | | 1.13 kB |
| RD-MIM-QSMS [15] | | 0.34 kB | 0.23 kB | 0.21 kB | 1.91 kB |
| RC-MIM-QSMS | | | 0.19 kB | | 1.32 kB |

## 6. Conclusions

In this paper, we proposed a framework of the RC-MIM-QSMS decoder to achieve less memory requirement for decoding QC-LDPC codes with different code rates. More specifically, we proposed the SMIM-DE to design the LUTs used by the RC-MIM-QSMS decoder, which took the weighted expectation of the pmfs and the joint degree distributions of RC-QC-LDPC codes into consideration. In such manner, we generated LUTs that varied with different layers and iterations. An optimization method was further adopted to unify the constructed LUTs into a unique set of iteration-specific LUTs. Simulation results showed that the proposed RC-MIM-QSMS decoder reduced memory usage by 93.22% compared to that of the RC-FAID decoder and achieved almost the same convergence speed compared to the RD-MIM-QSMS decoder. More importantly, the RC-MIM-QSMS decoder showed only minor performance degradation with respect to the RD-MIM-QSMS decoder and even slightly outperformed the floating-point SBP decoder in the high SNR region.

**Author Contributions:** Conceptualization, P.K. and X.H.; methodology, P.K. and X.H.; software, P.K.; validation, P.K., X.H. and K.C.; formal analysis, P.K. and X.H.; investigation, P.K. and X.H.; resources, P.K. and K.C.; data curation, P.K.; writing—original draft preparation, P.K.; writing—review and editing, P.K., X.H., and K.C.; visualization, P.K. and X.H.; supervision, K.C.; project administration, K.C.; funding acquisition, K.C. All authors have read and agreed to the published version of the manuscript.

**Funding:** This work was supported by the Singapore Ministry of Education Academic Research Fund Tier 2 T2EP50221-0036, by RIE2020 Advanced Manufacturing and Engineering (AME) programmatic grant A18A6b0057, by Fundamental Research Funds for the Central Universities under grant 2682022CX023, and by Natural Science Foundation of Sichuan under grant 2022NSFSC0952.

**Data Availability Statement:** The data presented in this study will be available on request from the corresponding author.

**Acknowledgments:** The authors would like to thank Singapore Ministry of Education, Agency for Science, Technology and Research (A*STAR), Ministry of Education of the People's Republic of China, and Natural Science Foundation of Sichuan for supporting this work.

**Conflicts of Interest:** The authors declare no conflict of interest.

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
