# Peer review of "Design of Mutual-Information-Maximizing Quantized Shuffled Min-Sum Decoder for Rate-Compatible Quasi-Cyclic LDPC Codes"

_electronics, doi:10.3390/electronics11193206_

Round 1
Reviewer 1 Report
The paper under review describes FAID decoders designed for rate-compatible quasi-cyclic LDPC codes. The authors proposed an optimization procedure that can generate the quantized decoder applicable to irregular codes with different code rates. The paper is up to date, well written and easy to follow. However, in my view the current version of the paper does not contribute enough beyond the state-of-the-art and I cannot recommend its publication, unless the manuscript is extended with some new knowledge and insights. My reasons are the following:
1. - The proposed optimization method, used to design proposed RC-MIM-QSMS decoders, combines the two approaches already published by the authors, given in references [14] and [15]. In [14] the authors proposed the rate-compatible optimization for flooding schedule, while in [15] single code rate MIM-QSMS decoder with layered schedule is proposed. It looks to me that merging the aforementioned methods is straightforward and, as such, more suitable for a conference paper, rather than the journal paper.
2. - The authors did not go much further beyond just proposing the decoder optimization method and verifying its applicability on one code set (with three code rates), which all in all provides rather limited contribution. Deeper insights are rarely provided. For example, some illustration of the optimization process is missing, together with comparison of density-evolution thresholds for different decoders given in Fig. 2 (this is of the major importance given that density evolution is used), analysis of different quantization levels and same insight related to the number code rates that can be optimized jointly (there probably must be a limit). Some validation that show relevance on the propose optimization is also mandatory: for example is type of the schedule important (comparison of the decoder designed for layered scheduled with decoder designed for the flooding schedule), or how important is to change LUTs in each iteration.
3. - Given the fact that rate-compatible codes are analyzed, it is preferable to show usefulness of the proposed decoder on LDPC codes defined in 5G NR standard.
4. - The authors noticed an interesting phenomenon: they reduced the error-floor compared to the belief-propagation decoder. This is interesting given the fact the error-floor optimization is not under the scope of the proposed optimization. Nevertheless, the authors should put an effort into explaining it. For example, is LUTs variation across decoding iterations reason for it, or does the phenomenon hold for other codes (for example 5G NR codes).
Reviewer 2 Report
This research is devoted to the optimization method of the lookup tables (LUT) for the rate-compatible mutual information-maximizing quantized shuffled min-sum (RC-MIM-QSMS) decoder, that helps to reduce the memory requirements for implementation preserving comparable convergence speed to the other state-of-the-art finite alphabet iterative decoder (FAIDs). The paper is good written. The result is clear.
The only thing that worries is that you say that in the paper [14]:
Kang, P.; Cai, K.; He, X.; Yuan, J. Memory Efficient Mutual Information-Maximizing Quantized Min-Sum Decoding for Rate Compatible LDPC Codes. IEEE Commun. Lett., 2022, 26, 733–737.
URL: https://ieeexplore.ieee.org/document/9684364
you previously consider the decoder with the flooding schedule and the main novelty of current paper is considering the decoder with the shuffled schedule. The problem is that the simulation results for proposed decoders represented in both papers are almost the same (I belive that they coincide).
Could you please comment this phenomenon or represent the comparison of this simulation results to see that they are different?
Round 2
Reviewer 1 Report
All my comments have been treated adequatly and I can recommend the publication of the manuscript in the currect form.
Reviewer 2 Report
This research is devoted to the optimization method of the lookup tables (LUT) for the rate-compatible mutual information-maximizing quantized shuffled min-sum (RC-MIM-QSMS) decoder, that helps to reduce the memory requirements for implementation preserving comparable convergence speed to the other state-of-the-art finite alphabet iterative decoder (FAIDs). The paper is good written. The result is clear.